The β-carboline alkaloid harmine inhibits telomerase activity of MCF-7 cells by down-regulating hTERT mRNA expression accompanied by an accelerated senescent phenotype

Zhao Lei 1 2
Wink Michael 2 wink@uni-hd.de
1 Department of Molecular and Experimental Medicine, Scripps Research Institute , La Jolla, CA , USA
2 Institute of Pharmacy and Molecular Biotechnology, Heidelberg University , Heidelberg , Germany
Kostyukova Alla
Electronic publication date: 2013 Oct 1
Publication date: 2013
Volume: 1
Electronic Location ID: e174
Received 2013 May 31; Accepted 2013 Sep 12
Copyright: © 2013 Zhao and Wink
Copyright year: 2013
Copyright holder: Zhao and Wink
License: This is an open access article distributed under the terms of the Creative Commons Attribution License, which permits unrestricted use, distribution, and reproduction in any medium, provided the original author and source are credited.
License URL: https://creativecommons.org/licenses/by/3.0/

Keywords: Telomerase, Senescence, MCF-7, p53, p21, Alkaloid, DNA intercalation, Apoptosis

Funding: The authors received no external funding for this study.

==============================
The end replication problem, which occurs in normal somatic cells inducing replicative senescence, is solved in most cancer cells by activating telomerase. The activity of telomerase is highly associated with carcinogenesis which makes the enzyme an attractive biomarker in cancer diagnosis and treatment. The indole alkaloid harmine has multiple pharmacological properties including DNA intercalation which can lead to frame shift mutations. In this study, harmine was applied to human breast cancer MCF-7 cells. Its activity towards telomerase was analyzed by utilizing the telomeric repeat amplification protocol (TRAP). Our data indicate that harmine exhibits a pronounced cytotoxicity and induces an anti-proliferation state in MCF-7 cells which is accompanied by a significant inhibition of telomerase activity and an induction of an accelerated senescence phenotype by over-expressing elements of the p53/p21 pathway.

Introduction

The end replication problem results in a continuous shortening of each end of a chromosome. In most somatic cells the shortened fragments cannot be compensated. Cells stop dividing when telomeres reach a critical length and replicative senescence is initiated consequently. However, most cancer cells can conquer this obstacle because their telomerase, a ribonucleoprotein that replicates telomere sequences at each cell division, remains active. Telomerase is highly associated with carcinogenesis. It is detectable in 85–90% of human cancers and over 70% of immortalized human cell lines (Kim et al., 1994; Shay & Wright, 1996a; Shay & Wright, 1996b), whereas it is undetectable in non-transformed somatic cells. Therefore, telomerase is an attractive target for the development of anti-cancer drugs.

Telomerase is a cellular reverse transcriptase containing two components: A protein element, telomerase reverse transcriptase (in human, hTERT) serving as catalytic subunit and an RNA element, hTR, providing a template for telomere synthesis (Nakamura et al., 1997).

Recent evidence suggests that increased telomere dysfunction leads to a loss of chromosome end protection and induces the senescence state. But senescence can also be induced without continuous telomere shortening suggesting that telomere integrity is critical regardless of telomere length. Tumour suppressor proteins such as p53 are required for the senescence arrest (Liu & Kulesz-Martin, 2001; Gorbunova, Seluanov & Pereira-Smith, 2002; Gewirtz, Holt & Elmore, 2008).

Harmine, a naturally occurring β-carboline alkaloid, has long been used in folk medicine in the Middle East and in Asia (Sourkes, 1999) and as a hallucinogenic drug (Wink & van Wyk, 2008). It was first isolated from the seeds of Peganum harmala L. in 1874 (Budavari, 1989; Roberts & Wink, 1998; Wink & van Wyk, 2008; Wink, Schmeller & Latz-Brüning, 1998). Harmine has multiple pharmacological properties including antiplasmodial activity (Astulla et al., 2008), antimutagenic and antiplatelet properties (Im et al., 2009). In vitro studies demonstrate that the planar structure of harmine leads to DNA intercalation. Since DNA intercalation causes frame shift mutations, these alkaloids are known to be mutagenic, cytotoxic and antimicrobial (Roberts & Wink, 1998; Wink, 2007). Burger and colleagues observed decades ago that harmine could inhibit monoamine oxidase (Burger & Nara, 1965) through which the metabolism of neurotransmitters are modulated (Kim, Sablin & Ramsay, 1997; Wink, Schmeller & Latz-Brüning, 1998). Recent data indicate that harmine and related alkaloids act as agonists at serotonin receptors (Wink, Schmeller & Latz-Brüning, 1998; Glennon et al., 2000; Song et al., 2004). Harmine and other β-carboline alkaloids therefore exhibit hallucinogenic properties (Wink & van Wyk, 2008).

Data obtained from cell viability assays indicate that harmine is a promising inhibitor of cell proliferation in a variety of cancer cell lines (Song et al., 2004). It blocks the cell cycle at G0/G1 phase (Hamsa & Kuttan, 2011) accompanied with a decrease of cyclin-dependent kinase activity (Song et al., 2002; Song et al., 2004). DNA intercalation is also involved in the inhibition of cell division as it prevents the transcription of several genes and causes frame shift mutations. Previous findings indicate that telomeres are also a target of intercalating drugs (Shammas et al., 2003; Shammas et al., 2004); they can induce very stable G-quadruplex structures which cannot be replicated by telomerase (Burge et al., 2006). Some known anticancer drugs exhibit DNA intercalation, such as isoquinoline, quinoline, and indole alkaloids (Wink, Schmeller & Latz-Brüning, 1998; Wink, 2007).

Among these alkaloids, the simple indole alkaloid harmine was identified in our laboratory as a potent DNA intercalating and cytotoxic natural product (Rosenkranz & Wink, 2007). It has been reported that a few DNA-intercalating alkaloids, including berbamine (Ji et al., 2002), chelidonine (Noureini & Wink, 2009) and 9-hydroxyellipticine (Sasaki et al., 1992) are inhibitors of telomerase activity. Because harmine is an intercalating and cytotoxic alkaloid a possible telomerase inhibition was evaluated in this research. The aim of this research was to examine the effects of harmine in human breast cancer MCF-7 cells and its possible interaction with telomerase. Anti-telomerase activity was analyzed using the telomeric repeat amplification protocol (TRAP). Harmine causes a pronounced cytotoxicity and induces an anti-proliferation state in MCF-7 cells. This process is accompanied by a significant inhibition of telomerase activity and an induction of an accelerated senescence phenotype by over-expressing p53/p21.

Materials & Methods

Chemicals

Harmine (C13H12N2O; MW 212.25) was purchased from Sigma-Aldrich. The stock solution was prepared in dimethyl sulfoxide (DMSO) with a concentration of 100 µM, which was stored at −20°C.

Cell culture and harmine treatment

Human breast cancer cell line MCF-7 was kindly provided by Prof. Dr. S. Wölfl (IPMB, Heidelberg University). Cells were routinely cultured in Dulbeccos’s Modified Eagle’s Medium (DMEM, Invitrogen) supplemented with 2 mM glutamine, 100 U/ml penicillin, 100 µg/ml streptomycin (Invitrogen, USA), and 10% heat-inactivated fetal bovine serum. Cells were incubated at 37°C in 5% CO2 and 100% humidity. Twenty-four h after plating, cells were treated with harmine and incubated up to different time points depending on the experimental design. A DMSO control was included in each analysis.

Metabolic cell activity assay

Ten microliters of 3-[4,5-dimethylthiazol-2-yl]-2,5-diphenyl tetrazolium bromide (MTT) (5 mg/ml) prepared in phosphate-buffered saline buffer were added to each well after given time intervals; all plates were gently shaken by hands for several times and incubated at 37°C for 3–4 h. At the end of incubation, the solution with MTT was carefully removed and 100 µl of lysis buffer (20% SDS in 1:1 N,N′-dimethylformamide: water/2% acetic acid/2.5% HCl 1 M) was added per well. Then the plates were placed on a shaker at low speed for 30 min at room temperature to ensure that the formazan formed was completely solubilized; it was quantified by measuring the OD value at 570 nm in a 96-well plate reader (Spectramax 384 plus, Molecular Devices).

Telomerase activity assay

Proteins were isolated from MCF-7 cells with CHAPS lysis buffer (10 mM Tris-HCl, pH 7.5; 1 mM MgCl2, 1 mM EDTA, 0.5% CHAPS, and 10% glycerol). All buffers and solutions were prepared with RNase-free water. Telomerase activities was determined with 0.5 µg protein extract using TRAP as described previously (Kim et al., 1994). Briefly, the protein extract was firstly incubated with TS primer (5′ AATCCGTCGAGCAGAGTT 3′) for 30 min at 30°C, after addition of CX primer (5′ AATCCCATTCCCATTCCCATTCCC 3′) the products were then subjected to PCR-amplification at 94°C for 30 s, and at 60°C 30 s for 29 cycles. The PCR products were separated on a 12.5% polyacrylamide gel by PAGE. The gel was stained with SYBR green (Amersham Biosciences) and directly visualized under a UV-transilluminator. A 36-bp internal control was amplified to serve as a standard for the normalization of telomerase expression. The intensity of all bands were photoscanned using ImageJ software (National Institutes of Health, America), the relative telomerase activity (RTA) was determined by the formula (Betts & King, 1999). RTA=(s−b)/ics(pc−b)/icpc×100

s: Intensity of sample

pc: Intensity of positive control

b: Intensity of background

ic: Intensity of internal control

Values are expressed as % of the control sample.

Reverse transcription PCR of endogenous hTERT expression

Total RNA was isolated from MCF-7 cells using RNeasy kit (Qiagen, Germany). One µg of total RNA was reverse transcribed in a 20 µL reaction volume using ImProm-II™ Reverse Transcription System (Promega, Germany). A 1 µL aliquot of cDNA was analyzed by PCR amplifications. Global hTERT was amplified using the primer 5′- CGGAAGAGTGTCTGGAGCAA-3′ paired with 5′-GGATGAAGCGGAGTCTGGA-3′; variant-hTERT was amplified with the primer 5′-GCCTCAGCTGTACTTTGTCAA-3′ paired with 5′-CGCAAACAGCTTGTTCTCCATGTC-3′. The thermocycling conditions for global hTERT amplification were: 94°C 2 min followed by 33 cycles of 94°C for 45 s, 60°C for 45 s, and 72°C for 90 s; for variant hTERT amplification, the thermocycling conditions were: 94°C for 2 min followed by 35 cycles of 94°C for 15 s, 60°C for 15 s, and 72°C for 30 s. The housekeeping gene β-actin was amplified with the primer 5′-CCTGGCACCCAGCACAAT-3′ paired with 5′-GGGCCGGACTCGTCATAC-3′ under the same thermocycling conditions described above with only 20 cycles. Amplified products (global hTERT: 145-bp; variant hTERT: full length variant, 457-bp; α variant, 421-bp; and β variant 275-bp; β-actin: 143-bp) were separated by gel electrophoresis on a 2% agarose gel and visualized by ethidium bromide staining.

Semi-quantitative PCR analysis

One microliter of cDNA was applied in 10 µL PCR reaction in capillaries containing 1 × SY BR Green Master Mix (ABgene), 0.3 µM of each primer. A non-template control was included as the negative control. The PCR reaction was performed in LightCycler3 (Roche, Germany) with initial 10 min denaturation at 95°C, then followed with 45 cycles: 95°C 10 s; 60°C 10 s. All crossing point (cp) values were assessed by using REST software relative to the expression of β-actin. Primers which were used in Real-Time PCR are listed in Table 1.

Table 1 Primers for RT-PCR.

Gene	Primer sequence (5′–3′)	
β-actin-f	CCTGGCACCCAGCACAAT	
β-actin-r	GGGCCGGACTCGTCATAC	
2007hTERT-f	ACGGCGACATGGAGAACAA	
2007hTERT-r	CACTGTCTTCCGCAAGTTCAC	
p21-f	TTTCTCTCGGCTCCCCATGT	
p21-r	GCTGTATATTCAGCATTGTGGG	
Cdk2-f	CCTCCTGGGCTGCAAATA	
Cdk2-r	CAGAATCTCCAGGGAATAGGG	
p53-f	TGCGTGTGGAGTATTTGGATG	
p53-r	TGGTACAGTCAGAGCCAACCAG	

β-galactosidase staining

MCF-7 cells were incubated with harmine for 48 h or 96 h before β-galactosidase activity was determined. Then cells were washed twice in PBS and fixed in fixation solution containing 2% formaldehyde and 0.2% glutaraldehyde for 5 min. The fixation solution was removed by washing the cells twice in PBS, and then the staining solution was added. Cells were then incubated at 37°C in a CO2 free environment for 8 h. The percentage of positively stained cells was determined after counting three random fields of 100 cells each. Representative microscopic fields were photographed under a 20× objective.

Western blot analysis for p53 and p21waf-1 proteins

MCF-7 cells were treated with 20 µM harmine for multiple time points (12, 24, 48, and 96 h) prior to lysing the cells in Nonidet-P40 (NP40) lysis buffer (20 mM Tris, pH 7.4, 150 mM NaCl, 5 mM EDTA, 1% NP40, and 10% glycerol). The constitutive levels of p53 and p21waf1 were assessed with respect to isogenic untreated MCF-7 cultures. Protein concentration was firstly determined with standard Bradford assay (Bradford, 1976), then a 25 µg aliquot of the protein extract was separated on 12% of SDS-PAGE and transferred onto a PVDF membrane (Millipore, Germany) by electroblotting. A standard blotting protocol was then performed using p53 (DO1; Santa Cruz Biotech, Germany) and a p21waf−1 monoclonal (BD Biosciences, Germany) antibody followed by horseradish peroxidase-conjugated anti-mouse IgG (Dianova GmbH, Germany). An enhanced chemiluminescent reaction (ECL Reagent, Amersham) was applied for the detection.

Results

Harmine is cytotoxic to MCF-7 cells in a dose- and time-dependent manner and induces accelerated senescence

The cytotoxicity of harmine in MCF-7 cells is shown in Fig. 1. Cell viability at various time points was determined by MTT assay. The results indicate that harmine arrests cell growth in a dose- and time-dependent manner. Concentrations of 20 and 30 µM harmine significantly reduced cell growth after 48 to 96 h. Concentrations between 10 and 20 µM did not influence viability of MCF-7 cells within the first 24 h, and were therefore used in the subsequent experiments.

Figure 1 Cytotoxicity of harmine in MCF-7 cells.

MCF-7 cells were incubated with harmine at different concentrations (0 µM, 10 µM, 20 µM and 30 µM) and multiple time periods (24 h, 48 h, 72 h and 96 h). Cell metabolic activity was determined by MTT assay. Viability of vehicle-treated samples was set at 100%: 24 h, white bars; 48 h, black bars; 72 h, hatched bars; 96 h, dotted bars. Results are derived from two independent experiments performed in quadruplicate (mean ± SD).

In the next step we tried to study whether senescent cells could be identified in response to harmine treatment. In MCF-7 cells, harmine arrests cell proliferation and induces a senescence morphology. β-Galactosidase activity, as a senescence marker, was detectable as early as 2 d after treatment with harmine, and became intense and expressed in virtually every cell of the culture at day 4 (Fig. 2). Cells, which were β-galactosidase positive, were larger in size or multinucleated (indicated with arrows), both of which are morphological features indicative of a senescent state. The SA-β-gal staining was not detected or barely detected in untreated control cells.

Figure 2 Harmine-induced senescence: SA-β-gal staining image of MCF-7 cells after harmine treatment.

MCF-7 cells were firstly treated with 20 µM harmine for 48 h and 96 h, respectively. At the end of treatment, SA-β-gal staining was investigated following a standard protocol. All images were taken at 10 × magnification. Percentage of β-gal positive cells were quantified by ImageJ software. Graph established from two independent areas (mean ± S.D). p values indicate the significant difference in positive β-gal staining for the sample treated with harmine with respect to the vehicle treated controls. Unpaired t test: ∗p ≤ 0.05; ∗∗∗p ≤ 0.001.

Telomerase activity

Telomerase activity of MCF-7 cells, treated with or without harmine, was evaluated as evidenced by the TRAP assay. A decreased telomerase activity (Fig. 3A) was detected after the incubation of the cells with 20 µM harmine. Telomerase activity was inhibited by 81.87% after 96 h treatment as compared to the untreated control (Fig. 3B). Treatment at a lower concentration, e.g., 10 µM, did not show a significant reduction of telomerase activity.

Figure 3 Effect of harmine on telomerase activity in MCF-7 cells.

(A) MCF-7 cells were incubated with harmine at two concentrations (10 µM and 20 µM) for 24 h, 48 h, 72 h and 96 h. At the end of incubation, telomerase activity was evaluated by applying TRAP assay; the TRAP products were then separated on a 12% PAGE gel and their intensity (all bands) was quantified by using ImageJ software and values were plotted in (B): ctr, vehicle control, black bar; cells treated with 10 µM of harmine, white bars; cells treated with 20 µM of harmine, dotted bars. Results derived from two independent experiments (mean ± SD). p values indicate the significant changes in relative telomerase activity for the sample treated with harmine with respect to the vehicle treated controls. Unpaired t test: ∗p < 0.05; ∗∗p ≤ 0.01.

Expression analysis of human TERT splicing variants by RT-PCR

RT-PCR analysis was performed with a pair of primers which covers all hTERT transcripts. In theory, four hTERT variants should be expected under the identical PCR conditions at the same time (full length hTERT with 457 bp; α variant with 421 bp; β variant with 275 bp and α + β variant with 239 bp). However, in our investigation, the α + β variant could not be detected (Fig. 4A). Treatment of the cells with 20 µM harmine significantly down-regulated all hTERT subunits in a time-dependent manner (Fig. 4B).

Figure 4 Harmine inhibits telomerase expression in MCF-7 cells in a dose- and time-dependent manner.

(A) MCF-7 cells were incubated with harmine at final concentration of 10 µM and 20 µM, respectively, then total RNA was isolated and analyzed by using RT-PCR; (B) MCF-7 cells were incubated with harmine at a final concentration of 20 µM for 48 h or 96 h, respectively, then total RNA was isolated and analyzed by RT-PCR.

Expression analysis of human TERT

Human hTERT, p21, and CDK2 mRNA transcripts were examined by real time PCR. Data were analyzed by Relative Expression Software Tool (REST2008). PCR efficiency was set as 2 as indicated by the software and the housekeeping gene β-actin was regarded as a control. A significant up-regulation of p21 mRNA was detected as early as 12 h after harmine treatment. The mRNA concentration was 3.9 fold higher than that of the untreated control, and the up-regulation became 6.5 fold with respect to the control after 96 h (Fig. 5). Within the first 24 h of treatment, no alteration of hTERT and CDK2 mRNA expression was detected, while an extended treatment up to 48 h showed that a significant down-regulation was observed for these two genes.

Figure 5 mRNA levels of hTERT, p21, and CDK2 in response to harmine treatment.

MCF-7 cells were exposed to harmine at a final concentration of 20 µM for 12 h, 24 h, 48 h and 96 h, then mRNA expression of each target gene was analyzed by real time PCR: hTERT, white bars; p21, black bars; CDK2, hatched bars. Results are representative of two independent experiments in triplicate (mean ± SD). p values measure significant changes in mRNA expression for the target gene treated with harmine with respect to the vehicle treated controls. Unpaired t test: ∗p > 0.05; ∗∗p ≤ 0.01; ∗∗∗p ≤ 0.001.

Harmine induces an over-expression of p53 and of p21

We had shown before that harmine arrested MCF-7 cell growth and induced senescence (Figs. 1 and 2). In order to define the mechanism of harmine-induced cell arrest a series of immunoblot analyses were performed (Fig. 6A). MCF-7 cells were cultured with harmine in a final concentration of 20 µM and cell samples were collected at different time points (24, 48, and 96 h). An enhanced expression of the phosphorylated H2AX (γH2AX) protein was detected after harmine treatment (Fig. 6B). An overexpressed p53 protein was identified by immunoblot analysis as early as 24 h accompanied by an increased p21 protein. c-Myc is a known transcriptional enhancer of hTERT expression. In our investigation, c-Myc was apparently down-regulated (Fig. 6B). Compared with the changes in other genes, the decrease of c-Myc was more moderate in response to the treatment with harmine.

Figure 6 Harmine induces a general DNA damage response byover-expressing p53/p21 and γH2AX.

(A) MCF-7 cells were incubated with harmine at 20 µM for 24 h, 48 h and 96 h, then 25 µg of total protein extracted from cells after treatment of harmine or vehicle only was separated by PAGE and analyzed by Western blot. (B) changes in protein level after the treatment were calculated with respect to vehicle controls (100%): p21, black bars; p53, white bars; γH2AX, grey bars; c-Myc, hatched bars.

Discussion

The indole alkaloid harmine exhibits multiple pharmacological properties in vivo and in vitro (Wink & van Wyk, 2008; Wink & Schimmer, 2010). Among other effects, harmine significantly arrests cell proliferation and induces cell death in a number of tumour cell lines. A dose- and time-dependent cytotoxicity of harmine could be confirmed in our experiments with MCF-7 cells (Fig. 1).

Cytotoxicity can result from an adverse interaction of harmine and other alkaloids with one or more important targets present in a cell including DNA, RNA, or associated enzymes (Roberts & Wink, 1998; Wink & van Wyk, 2008; Wink, 2007). Harmine is known to intercalate DNA and can cause mutations and DNA damage (Wink, Schmeller & Latz-Brüning, 1998; Wink & Schimmer, 2010). Through these interactions cell proliferation can be interrupted or cell death induced (Lansiaux et al., 2002; Möller et al., 2007; Wink, 2007). In addition, the inhibition of cycline-dependent kinases such as CDK2 and CDK5 (Song et al., 2002) might also contribute to the cytotoxicity of harmine. Furthermore, it has been shown that harmine can repress cytochrome P450 activity (Tweedie, Prough & Burke, 1988) and selectively inhibit DNA topoisomerase (Funayama et al., 1996).

Another mechanism for cytotoxicity of alkaloids might involve the intercalation of telomeres and the inhibition of telomerase. Several DNA-intercalating alkaloids, including berbamine (Ji et al., 2002), chelidonine (Noureini & Wink, 2009) and 9-hydroxyellipticine (Sasaki et al., 1992) could significantly inhibit telomerase activity which could lead to the interruption of the genomic stability as well as cell growth arrest (Shay & Bacchetti, 1997). Because harmine is an intercalating alkaloid a possible telomerase inhibition was evaluated in this research. Indeed, as shown in our investigation, harmine induces a remarkable reduction of telomerase activity in MCF-7 cells as measured by TRAP (a PCR-based assay to detect telomerase activity) (Fig. 3). Harmine also triggers a significant inhibition of telomerase activity in Hela cells (Zhao, 2010), the concentrations applied in both cell lines were very similar (20 µM in MCF-7 cells, 30 µM in HeLa cells). Under our experimental condition, we did not find the same senescent phenotype with HeLa cells. Also no down-regulation could be detected on hTERT mRNA expression although telomerase activity was significantly inhibited after harmine treatment. The mTOR pathway might be involved and needs to be further investigated. The regulation of telomerase activity involves various signalling pathways (Shay & Wright, 1996a; Shay & Wright, 1996b). It is commonly accepted that the expression of hTERT is critical for telomerase activity (Meyerson et al., 1997; Nakamura et al., 1997; Bodnar et al., 1998). The transcription of hTERT mRNA was significantly down-regulated in response to harmine treatment (Figs. 4 and 5). The down-regulation became apparent about 24 h earlier than the reduction of telomerase activity (TRAP assay) whereas no decrease in telomerase activity could be seen at the same condition. Such an observation coincides with the report that telomerase activity has a half-life longer than 24 h in almost all cell lines (Holt et al., 1997) whereas the half-life of the hTERT mRNA is about 2 h (Xu et al., 1999). Our hypothesis is harmine does not induce a direct inhibition on telomerase activity in MCF-7 cells but through down-regulating hTERT at transcriptional level.

Another factor could be c-Myc which plays an important role in the transcriptional regulation of hTERT (Wu et al., 1999; Kyo et al., 2000). Overexpressed c-Myc protein leads to a remarkable E-box dependent increase in the hTERT promoter activity. Moreover, c-Myc could induce the expression of endogenous hTERT mRNA and telomerase activity in normal human cells (Wang et al., 1998; Greenberg et al., 1999). In our experiments, a time-dependent down-regulation of cMyc was observed (Fig. 6) which might be correlated with the down-regulation of hTERT (Fig. 4).

The tumor suppressor protein p53 is a nuclear transcription factor that accumulates in response to cellular stress, including DNA damage and oncogene activation (Wink, 2007). P53 protein is a critical determinant of the cell fate following certain types of DNA damage (Clarke et al., 1993; Liu & Kulesz-Martin, 2001). DNA damage triggers transcriptional transactivation of p53 target genes such as p21, leading to cell cycle arrest, senescence and/or apoptosis (Levine, 1997; Farnebo, Bykov & Wiman, 2010). P53 is essential for both senescence and apoptosis pathways, specifically, in cell cycle arrest at G1 phase; p53 enhances p21 transcription, which in turn inhibits CDK activity. As reported, overexpressed wild-type p53 could inhibit telomerase activity via down-regulating hTERT transcription (Gollahon et al., 1998; Kusumoto et al., 1999). However, such a reduction cannot be directly achieved by p53 because the binding site between p53 and the hTERT promoter is missing (Gualberto & Baldwin, 1995; Bargonetti et al., 1997; Kyo et al., 2000).

In our study, p53 was overexpressed after harmine exposure (Fig. 6); an induction could be detected as early as 12 h after treatment. This enhancement was accompanied by an increase in mRNA level as well as on protein level of p21 (Figs. 5 and 6). The question arises as to whether the inhibition of hTERT is a consequence of overexpressed p53 or harmine-induced cell cycle arrest. Harmine is able to interrupt DNA replication in vivo (Boeira, Erdtmann & Henriques, 2001; Moura et al., 2007; Sasaki et al., 1992) and in vitro (Wink, 2007). Other studies have found that harmine induces chromosome aberrations and produces DNA breakage in cultured mammalian cells (Boeira, Erdtmann & Henriques, 2001). Moreover, harmine can inhibit topoisomerase I (Cao et al., 2005; Wink, Schmeller & Latz-Brüning, 1998), therefore blocking an important enzyme which can repair DNA damage and fix mutations (Sasaki et al., 1992; Wang, 1998). The accumulation of phosphorylated H2AX (γH2AX) is an early sign of genomic events reflecting induction of double strands breaks (Tanaka et al., 2007; Albino et al., 2004). In this study, an increase of γH2AX was detected at 24 h after the treatment of harmine. Our hypothesis is that intercalating harmine induces a general time-dependent DNA damage response. Instead of triggering apoptosis, such damage apparently initiates an accelerated senescence in MCF cells (Fig. 2). Similar results were obtained in other studies, in which MCF-7 cells failed to undergo apoptotic cell death but underwent accelerated senescence after the exposure of ionizing radiation and adriamycin (Elmore et al., 2002; Jones et al., 2005). On the other hand, when p53 protein was diminished by infection with HPV-E6 oncogene, MCF-7-E6 cells entered delayed programmed cell death (Elmore et al., 2002). A number of studies have demonstrated that replicative senescence induced by telomere shortening and DNA damage-induced senescence leads to a very similar cell morphology (Oh et al., 2001; Gorbunova, Seluanov & Pereira-Smith, 2002; Gewirtz, Holt & Elmore, 2008). Both events involve the participation of p53, the mechanisms, however, remain unclear.

In conclusion, the treatment of MCF-7 cells with the DNA intercalator harmine induces a time-dependent general DNA damage response. P53 senses the damage and stops cell cycle progression by transactivating p21. Alternatively, the overexpressed p53 could directly inhibit hTERT transcription. The inhibited telomerase could then facilitate cell growth arrest in MCF-7 cells, and directs damaged cells into accelerated senescence and not into apoptotic pathway.

We thank Holger Schäfer for helpful discussion.

Additional Information and Declarations

Competing Interests

Author Contributions

Michael Wink is an Academic Editor for PeerJ.

Lei Zhao conceived and designed the experiments, performed the experiments, analyzed the data, wrote the paper.

Michael Wink conceived and designed the experiments, contributed reagents/materials/analysis tools, wrote the paper.

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
