# Peer review of "The β-carboline alkaloid harmine inhibits telomerase activity of MCF-7 cells by down-regulating hTERT mRNA expression accompanied by an accelerated senescent phenotype"

_PeerJ, doi:10.7717/peerj.174_

## Round 0.1 · original submission · Minor Revisions

The paper will be accepted if you address the reviewers comments.

Reviewer 1 ·

Basic reporting

The introduction needs to be improved as follows:

- may you add more references to the sentence (line 54): Telomeres are affected by intercalating drugs. It is indeed one of the main points for doing the work, so it needs to be more referred.

- in respect with this first comment, may you explain better the rational of doing such a study, why do you think intercalating drugs affects telomerase activity ? It’s obvious that intercalating drugs cause DNA damages and block cell cycle and cell proliferation. This may mean that telomerase substrate is affected as well. However, the link between such drugs and telomerase enzymatic activity itself is not trivial. This kind of argumentation is missing in the introduction part and needs to be developed. (surprisingly, it is well argued in the discussion part)

Experimental design

no comments

Validity of the findings

no comments

Additional comments

In the present manuscript, the authors explored more in depth the molecular mechanism through which harmine may exert its noxious effects on cancer cells and established that this natural alkaloid strongly induces senescence in MCF-7 cells. This action is accompanied by a loss of telomerase expression and activity, and correlates well with an induction of the p53/p21 signaling pathway.

This work is generally well done. Data are clear and fairly interpreted. The manuscript is well written. However, a few points still need to be improved:

Major comments:

1) the title is not totally correct in the sense that you did not prove that the senescence induced by harmine is due to p53/p21 induction or telomerase inhibition by using SiRNA or inhibitors. At this time, you can just say that these two last effects occur concomitantly to the appearance of the senescent phenotype. So may you change the title or provide more experiments to argue the data.

2) in the results section:

- may you add the % of cells -gal+ in each image of Figure 2 ?

- in Figure 4B, you show that all detectable telomerase variants strongly disappeared 48h post-harmine treatment, while in Figure 3, you only detect a small decrease (20%) of telomerase activity at the same time and concentration of drug treatment. May you clarify this point?

- the title of the last paragraph of the results (down-regulation of hTert…) does not really correlate with the data shown and needs to be changed

3) as a general discussion :

- it is generally admitted that MTT assays measure cell “cytotoxicity” but, this word also has a connotation of cell death. In your case, it rather seems that harmine stops cell proliferation and has a cytostatic effect towards MCF-7 than it really induces the death of the cells and harbour a cytotoxic effect, which you kind of explained well. Overall, MCF7 sensitivity to harmine was evaluating by MTT assay might be more appropriate.
As this assay really measures the metabolic state of your cells, and considering that senescent or arrested cells are less metabolically active than proliferating cells, is it possible to distinguish between the loss of metabolic activity due to cell proliferation inhibition over senescence? In other terms, in your assays are you measuring cell growth inhibition or a hallmark of senescence or both? is there any literature on that ?

- do your cells finally die ? or can they be maintained like that for a long time ?
- As harmine induced a loss of telomerase activity in HeLa, do you also have a senescent phenotype? If not, can it be related to mTor pathway activation ?


Minor comments: Some typing errors are present all along the manuscript:
- in the abstract, line 4, there is a space between h and armine
- in the introduction, line 26, there is : A protein … it may be : a protein
- in the mat and meth section, there is a space in the cell culture and harmine treatment paragraph
- in the discussion, sometimes p53 is written P53 and sometimes P53; check it… I guess it is p53 except if it starts the sentence and line 271, 7 is missing from MCF.

·

Basic reporting

The manuscript of Zhao and Wink deals with the senescence-inducing alkaloid harmine. The authors found that harmine upregulates signal transducers of the p53/p21 pathway and inhibit telomerase activity in MCF-7 cells.
This is an interesting manuscript which is well designed and performed. The results are novel and significant and the manuscript deserves publication from my point of view.

Experimental design

Two aspects should be considered, before the manuscript can be accepted:
1. A rationale should be given, why the authors specifically focus on harmine (and not another drug) for its telomerase-inhibiting activity.
2. The authors should include a paragraph on the potential of alkaloids or phytochemicals in general as telomerase inhibitors. Harmine could be embedded in a more generalized context to emphasize the relevance of harmine.

Validity of the findings

It is important to note that the harmine concentrations used for cytotoxicity testing were comparable to those used to investigate the molevular mode of action oif the compound. Hence, the observed data are meaningful.

Additional comments

To my point of view, the manuscript is acceptable after minor revision.

---

## Round 0.2 · Minor Revisions

Just few changes should be done before your paper is accepted (see reviewer's comments).

Reviewer 1 ·

Basic reporting

The authors replied to all the questions asked. The argumentation looks more convincing. The discussion part is really good. I believe that the paper is now ready for publication. I just have three more comments:

1- In the introduction:
a. Line 49, the word “also” is not at the right place in the sentence. It should be ‘also indicate’ or also ‘telomeres are also..”
b. Line 54 to 57: I still miss why you first made the hypothesis that Harmine could directly interact with or inhibit hTert. You may briefly repeat what you are actually explaining in the discussion part, line 213 to 218.


2- In the results section, I appreciate that you change the last title of the results, but I still don’t get why you say “ independent of p21 induction” . I feel like p21 is induced as much as p53, and it is difficult to say if it is dependent or not of p53.


3- In the Figures: Figure # 4 is labeled Figure # 1 instead of 4

Experimental design

No comments

Validity of the findings

No comments

---

## Round 0.3 · accepted · Accept

Congratulations with the accepted manuscript.